# DNA Methylation and Histone Acetylation Contribute to the Maintenance of LTP in the Withdrawal Behavior Interneurons in Terrestrial Snails

**DOI:** 10.3390/cells13221850

**Published:** 2024-11-08

**Authors:** Alena Zuzina, Daria Kolotova, Pavel Balaban

**Affiliations:** Cellular Neurobiology of Learning Laboratory, Institute of Higher Nervous Activity and Neurophysiology, Russian Academy of Sciences, 117485 Moscow, Russia; kolotova@ihna.ru

**Keywords:** epigenetics, DNA methylation/demethylation, histone acetylation, synaptic plasticity, gastropods

## Abstract

Accumulated data indicate that epigenetic regulations, including histone modifications and DNA methylation, are important means for adjusting the expression of genes in response to various stimuli. In contrast to the success in studying the role of DNA methylation in laboratory rodents, the role of DNA methylation in the terrestrial snail *Helix lucorum* has been studied only in behavioral experiments. This prompted us to further investigate the role of DNA methylation and the interaction between DNA methylation and histone acetylation in the mechanisms of neuroplasticity in terrestrial snails using in vitro experiments. Dysregulation of DNA methylation by the DNMT inhibitor RG108 significantly suppressed the long-term potentiation (LTP) of synaptic inputs in identified neurons. We then tested whether the RG108-induced weakening of potentiation can be reversed under co-application of histone deacetylase inhibitors sodium butyrate or trichostatin A. It was found that increased histone acetylation significantly compensated for RG108-induced LTP deficiency. These data bring important insights into the functional role of DNA methylation as an important regulatory mechanism and a necessary condition for the development and maintenance of long-term synaptic changes in withdrawal interneurons of terrestrial snails. Moreover, these results support the idea of the interaction of DNA methylation and histone acetylation in the epigenetic regulation of synaptic plasticity.

## 1. Introduction

DNA methylation is known to play important roles in the regulation of long-term synaptic plasticity, learning and memory [1,2,3,4] and is primarily associated with the suppression of gene expression [5,6,7]. DNA methylation is carried out by a family of enzymes known as DNA methyltransferases (DNMTs) and, as a rule, involves the covalent transfer of a methyl group to the C-5 position of the cytosine ring of DNA. Previous studies demonstrated that DNMT is involved in long-term potentiation (LTP) not only in vertebrates (in medial prefrontal cortex [4] and hippocampus [3]) but also in mollusks (in *Aplysia californica* [8,9]). Additionally, the DNMT involvement has also been documented previously for long-term memories [10,11,12,13,14]. Several studies discovered that training induced up-regulation of DNMT mRNA expression. Interestingly, all of those changes were found within 30–90 min after the training procedure. Indeed, DNMT3a and DNMT3b transcript levels were increased in the dentate gyrus following novel object-location learning [15]. Auditory fear conditioning led to increased DNMT3A mRNA expression in the lateral amygdala [16]. In animals exposed to context fear conditioning, an increase in DNMT3A and DNMT3B mRNA in the CA1 hippocampus region was observed [10]. After training in cued fear conditioning, the Dnmt3a mRNA rapidly increased in the amygdala [17]. In addition to changes in the DNMT mRNA levels, dynamic changes in DNA methylation have also been shown. Halder with colleagues [18] showed learning-induced DNA methylation changes 1 h after context fear conditioning. Changes in DNA methylome of dentate granule cells in the adult mouse hippocampus were observed 4 h after a single synchronous activation by electroconvulsive stimulation [19]. Although DNA methylation has been known to be involved in LTP [3], to date, very few studies have attempted to observe the scale of methylation changes in LTP. Maag with colleagues [20], using a methylated DNA immunoprecipitation array, discovered widespread changes in the methylation status of LTP genes: methylation changes were found at 30 min, 2 h, and 5 h post-high-frequency stimulation.

Histone acetylation is one of the best studied histone modifications. Histone deacetylases (HDACs) and histone acetylases control chromatin structure through deacetylation and acetylation, respectively, of the N-terminal histone lysine residues. The more acetylated chromatin is, the more relaxed it is, which in turn leads to an increase in gene transcription [21,22,23]. According to the modern point of view, histone acetylation is an essential regulatory mechanism of synaptic plasticity. A significant increase in histone acetylation was shown both in vitro and in vivo experiments following the induction of long-term plasticity changes. Acetylation of histone H3 was increased in vitro in the hippocampus and in vivo, the contextual fear conditioning was accompanied by similar increases in histone acetylation within the hippocampus 1 h after training [24]. Treatment with 5-HT increased the acetylation of both histone H3 and H4 at the C/EBP promoter 90 min after 5-HT treatment [25]. Danilova and Grinkevich [26] observed a significant increase in histone H3 acetylation in the subesophageal complex of ganglia in gastropod mollusk *Helix* 1 h after learning. Collectively, a relatively large body of evidence suggests that histone acetylation levels change in the first few hours following learning.

One of the methods of regulating the level of histone acetylation in experiments is pharmacological inhibition. Sodium butyrate (NaB) and trichostatin A (TSA) are widely used histone deacetylase inhibitors (HDACis); TSA is supposed to inhibit both class I and class II HDAC enzymes [27] and NaB inhibits only class I HDACs without affecting class IIa, Iib, or III HDACs [28]. Several studies showed that HDACi can ameliorate LTP deficit [16,29,30,31].

Terrestrial snail *Helix lucorum* provides researchers with a relatively simple nervous system that tolerates laboratory conditions well, with large neurons with strictly defined functions [32]. At the same time, terrestrial snails exhibit quite complex behaviors, which makes them suitable candidates for the study of neuroplasticity processes. Withdrawal interneurons in terrestrial snails (also known as command neurons for avoidance behavior) are involved in synaptic plasticity underlying avoidance behavior. The cellular mechanisms for this behavior and plasticity are well characterized [33]. However, the molecular pathways underlying them are not well understood. Recently, it was found that epigenetic modulation is a necessary regulatory component of the long-term synaptic plasticity and long-term memory in terrestrial snails [14,30,34,35,36]. The presence of DNMTs, their encoding genes and all the necessary machinery in mollusks is beyond doubt [37,38,39,40,41]. However, the contribution of DNA methylation was only studied in in vivo experiments using two models of conditioned reflex learning (food aversion and contextual avoidance memory) [14,34]. Disruption of DNA methylation by the DNA methyltransferase inhibitor (DNMTi) RG108 has been shown to impair maintenance of both types of memory. This prompted us to further investigate the role of DNA methylation and the interaction between DNA methylation and histone acetylation in the mechanisms of neuroplasticity in terrestrial snails using in vitro models and intracellular recording of synaptic events in functionally identified neurons.

In the present study, we tested our hypothesis that DNA methylation in cooperation with histone acetylation regulates synaptic plasticity by determining the effects of DNMTis and HDACis on LTP in withdrawal interneurons in terrestrial snail. We demonstrated that a session of five tetanizations associated with serotonin (5-HT) applications led to sustained long-lasting potentiation, and dysregulation of DNA methylation by DNMTi RG108 suppressed this LTP significantly. We then tested if co-application either of HDACis NaB or TSA with RG108 prevented the weakening of potentiation. We showed that increased histone acetylation significantly compensated for RG108-induced LTP deficiency. These data bring important insight to the functional role of DNA methylation as an important regulatory mechanism and a necessary condition for the development and maintenance of long-term synaptic changes in withdrawal interneurons in terrestrial snails.

## 2. Materials and Methods

### 2.1. Animals

The work was performed in adult *Helix lucorum taurica* L. All animals were kept in terrariums under laboratory conditions. The terrariums were provided with high humidity, a sufficient amount of food, water, and a low concentration of animals, which contributed to the normal active existence of animals. In all experiments, animals similar in weight and size (15 g ± 5) were used. Before the experiments, the snails were in an active state for at least two to three weeks. Experimental procedures were in compliance with the Guide for the Care and Use of Laboratory Animals published by the National Institutes of Health, and the protocol was approved by the Ethical Committee of the Institute of Higher Nervous Activity and Neurophysiology of Russian Academy of Sciences.

### 2.2. Electrophysiological Experiments

The work was carried out on the isolated central nervous system of terrestrial snails. Details of preparation have been reviewed and will not be detailed here [32]. Using sharp glass microelectrodes filled with 2 M potassium acetate (tip resistance, 15–20 MOhm), the activity of visually identified giant withdrawal premotor (command) interneurons (parietal #3 and parietal #2) of the parietal ganglia was recorded. Excitatory postsynaptic potentials (EPSPs) evoked by electrical stimulation (duration 3 ms) of the second cutaneal (glutamatergic-like synapses [42]) or intestinal (acetylcholinergic-like synapses [43]) nerve were recorded. Intracellular signals were recorded with preamplifiers (Axoclamp 2B, Axon Instruments, Union City, CA, USA), digitized, and stored on a computer (Digidata 1400A A/D converter and Axoscope 10.0 software, both from Axon Instruments, CA, USA).

In each experiment, stimulus intensity was adjusted to evoke EPSPs of 4–10 mV amplitude from each stimulated nerve in withdrawal interneurons. In control experiments aimed at studying basic synaptic plasticity, test stimulation of the second cutaneal or intestinal nerves was carried out throughout the recording session with an interval between stimuli of 10 min. To study the long-term plasticity, a homosynaptic potentiation protocol was used. At the beginning of the recording, five test stimulations of the second cutaneal or intestinal nerves were performed with an interval between stimuli of 10 min, and then tetanization of the second cutaneal or intestinal nerve was carried out (a burst of stimuli with a frequency of 10 Hz, a burst duration of 10 s, a 10-fold increase in the amplitude of the test stimulus). In total, tetanization trains were delivered five times with an interval of five minutes (Figure 1). Before each tetanization, 5-HT was added to the experimental bath, which was washed off two minutes after the tetanization. After the fifth tetanization, test stimulation of the second cutaneal or intestinal nerve was continued at the original stimulus amplitude every 10 min for at least 4 h.

### 2.3. Drugs

HDACi sodium butyrate (NaB) (Sigma, St. Louis, MO, USA) and serotonin (5-HT) (Tocris, Bristol, UK) were dissolved in sterile physiological Ringer solution (in mM: 100 NaCl, 4 KCl, 7 CaCl_2_, 5 MgCl_2_, and 10 Tris-HCl buffer (pH 7.8)). The DNMT inhibitors N-phthalyl-L-tryptophan (RG108) (Sigma, St. Louis, MO, USA) and trichostatin A (TSA) (Sigma, St. Louis, USA) were dissolved in DMSO (dimethyl sulfoxide) as stock solution. The final concentration of the substances in the experimental bath was as follows: for DMSO, the concentration was less than 0.1% (shown to be inactive in our preparations); for NaB, it was 6 × 10^−5^ M; for 5-HT, it was 10^−5^ M; for RG108, it was 20 × 10^−5^ M; for TSA, it was 0.5 × 10^−6^ M (these concentrations were selected based on literature data [14,30]). The drugs were added to the experimental bath using a pipette.

Depending on the experimental design, either RG108 or RG108+NaB or RG108+TSA were applied to the bath solution 40 min before the first tetanization. During the tetanization session, an intensive “washing out” of the drugs was carried out.

The experiments included the following groups: “Control” groups—the groups in which only test stimulation was carried out. Group “control+RG108”—RG108 was in the experimental bath for the first 40 min of recording. “LTP” groups—groups in which a homosynaptic protocol was used to produce LTP. “LTP+RG108” group—the LTP protocol was used and RG108 was in the experimental bath for the first 40 min of recording. “LTP+NaB” and “LTP+TSA”—these groups were characterized by LTP and simultaneous application of the DNMTi RG108 and HDACi (NaB or TSA).

### 2.4. Data Analysis

The significant differences in the amplitudes of EPSPs were assessed using the Mann–Whitney test. The standard program STATISTICA 10.0 was used to process the results. All data are presented as mean ± S.E.M. Differences were considered significant at *p* < 0.05, and were denoted in figures by *#@%.

## 3. Results

In an effort to investigate whether DNA methylation contributes to long-term potentiation (LTP) in withdrawal interneurons of the snail nervous system, we used DNMTi RG108. This inhibitor was shown to bind the active site of DNMT and to inhibit the enzyme by blocking the catalytic domain without the need for incorporation into DNA. In addition, RG108 is much less cytotoxic than nucleoside inhibitors 5-Aza-2′-deoxycytadine and zebularine [44]. Note that by stimulating either the second cutaneal nerve (glutamatergic-like synapses [42]) or the intestinal nerve (acetylcholinergic-like synapses [43]), we investigated the effect of DNMT inhibition on the plasticity of two synaptic inputs with different transmitters from sensory neurons to premotor interneurons. In the first series of experiments, test stimulation of the second cutaneal nerve was performed. In all control recordings without tetanization (groups: control, n = 10; control+RG108, n = 9), test stimulation led to a gradual decrease in EPSP amplitude (100% of the initial EPSP amplitude is shown at time point “−40” min; control: 50.2 ± 7.7% at time point 120 min, 34.1 ± 6.4% at time point 250 min; control+RG108: 58.2 ± 8.4% at the time point 120 min, 42.2 ± 5.8% at time point 250 min). No significant differences were found between the EPSP amplitude in the control and control+RG108 groups (Figure 2A). Thus, our findings suggest that RG108 did not affect the amplitudes of non-potentiated glutamatergic EPSPs in withdrawal interneurons.

Next, we determined whether the RG108 administration affected the LTP of synaptic inputs in withdrawal interneurons. During the first five test stimuli, the EPSP amplitude gradually decreased in both LTP groups (LTP, n = 10; LTP+RG108, n = 8). No differences were observed between EPSP amplitudes in the control, control+RG108, LTP and LTP+RG108 groups at the 0 min time point (Figure 2A,D). Five tetanizations combined with 5-HT applications caused a pronounced increase in EPSP amplitude in both LTP groups. Thus, immediately after the last tetanization combined with the application of 5-HT (30 min time point), the EPSP amplitude in the LTP group was 226.0 ± 32.1%, and in the LTP+RG108 group, it was 274.8 ± 28.0%. There were no significant differences between the LTP and LTP+RG108 groups at the 80 min time point, while between the LTP and control groups, significant differences were found (Figure 2A,E). The decrease in the EPSP amplitudes in the LTP+RG108 group compared to the LTP group began 90 min after the completion of tetanization. Thus, at the time point 120 min, the EPSP amplitude in the LTP group was 172.3 ± 16.3%, while in the LTP+RG108 group, it was only 105.4 ± 33.1% (*p* = 0.043089). In addition, 4 h after tetanization (time point 250 min), the EPSP amplitude in the LTP group significantly exceeded the amplitude of responses in the LTP+RG108 group (LTP—120.2 ± 14.4%; LTP+RG108—25.6 ± 4.4%, *p* = 0.000250). Moreover, no differences were observed between EPSP amplitudes in the control, control+RG108 and LTP+RG108 groups at the 250 min time point (Figure 2A,F). It should be noted that from the 120 min time point to 250 min time point of the recordings, a significant difference was observed between the LTP and LTP+RG108 groups (*p* < 0.05, Figure 2A), while the EPSP amplitudes of the LTP+RG108 group did not differ significantly from those of the control group during the last 100 min of recordings (Figure 2A). Thus, the RG108 administration 40 min before LTP induction did not block the initial phase of LTP in glutamatergic synapses, but led to a significant weakening of the late phase of LTP in withdrawal parietal interneurons.

To address possible involvement of histone acetylation and its interaction with DNA methylation in LTP of withdrawal interneurons, we investigated the role of HDAC by adding HDACis (NaB or TSA) to the bath solution together with RG108 (Figure 2, groups LTP+RG108+NaB, n = 8, LTP+RG108+TSA, n = 8). Test stimulation before the start of the potentiation protocol led to a gradual slight decrease in EPSP amplitude in LTP+RG108+NaB and LTP+RG108+TSA groups, as in the groups described earlier (control, control+RG108, LTP, LTP+RG108). No differences were observed between those groups at the 0 min time point (Figure 2B,D). Five tetanic stimulations combined with 5-HT applications caused a significant increase in the EPSP amplitude: the LTP of synaptic responses was observed in LTP+RG108+NaB and LTP+RG108+TSA groups (Figure 2B). Notably, the EPSP amplitudes in LTP+RG108+NaB and LTP+RG108+TSA were not affected by RG108 administration: EPSPs in LTP+RG108+NaB and LTP+RG108+TSA did not differ from those in the LTP group during the recording (Figure 2B,D–F). In addition, the EPSP amplitude in LTP+RG108+NaB and LTP+RG108+TSA significantly exceeded the EPSPs in LTP+RG108 for the 210–250 min and 200–250 min time windows, respectively (Figure 2B, Table 1). Thus, as HDACi (NaB or TSA) applications led to increased glutamatergic EPSP amplitudes in the late phase of LTP, it can be assumed that the increased level of histone acetylation can compensate for the RG108-induced LTP deficits.

In the second series of experiments, test stimulation of the intestinal nerve (putative acetylcholinergic synaptic inputs) was performed. In all control recordings without tetanization (groups: control, n = 10; control+RG108, n = 10), test stimulation led to a gradual decrease in EPSP amplitude (control: 58.0 ± 7.0% at time point 120 min; 32.3 ± 5.2% at time point 250 min; control+RG108: 45.4 ± 4.4% at the time point 120 min; 38.3 ± 6.0% at time point 250 min). No significant differences were found between the EPSP amplitudes in the control and control+RG108 groups (Figure 3A). Thus, the results suggest that RG108 applications do not affect the amplitude of non-potentiated acetylcholinergic EPSPs in withdrawal interneurons.

Next, we investigated whether RG108 affects the LTP induction and maintenance in acetylcholinergic synapses in terrestrial snails. During the first five test stimuli, the EPSP amplitude gradually decreased in both LTP groups (LTP, n = 8; LTP+RG108, n = 13). No differences were observed between EPSP amplitudes in the control, control+RG108, LTP and LTP+RG108 groups at the 0 min time point (Figure 3A,D). Five tetanizations combined with 5-HT applications caused a pronounced increase in EPSP amplitude in both LTP groups. Thus, immediately after the last tetanization combined with the application of 5-HT (30 min time point), the EPSP amplitudes in the LTP group were 187.0 ± 19.7%, and in the LTP+RG108 group, these were 168.1 ± 19.6%. However, despite the fact that the EPSP amplitudes in the LTP+RG108 group were not affected by RG108 for the first 50 min after the tetanization protocol started, they became significantly reduced by 80 min in comparison to the EPSP amplitudes in the LTP group (Figure 3A,C,E). Weakening of the EPSPs in the LTP+RG108 group relative to the LTP group continued during the rest of recording. Thus, at the time point 120 min, the mean EPSP amplitude in the LTP group was 178.5 ± 24.2%, while in the LTP+RG108 group, it was only 105.4 ± 15.8% (*p* = 0.043089). In addition, 4 h after tetanization (time point 250 min), the mean EPSP amplitude in the LTP group still significantly exceeded the amplitude of responses in the LTP+RG108 group (LTP—152.3 ± 15.3%; LTP+RG108—47.1 ± 17.5%, *p* = 0.000250). Moreover, no differences were observed between EPSP amplitudes in the control, control+RG108 and LTP+RG108 groups at the 250 min time point (Figure 3A,F). It should be noted that from the 110 min time point to the 250 min time point of the recordings, a significant difference was observed between values in the LTP and LTP+RG108 groups (*p* < 0.05, Figure 3A), while the EPSP amplitudes in the LTP+RG108 group did not differ significantly from those of the control group during the last 130 min of recording (Figure 3A). Thus, the RG108 administration 40 min before LTP induction weakened the initial phase of LTP, and also led to a significant decrease in the late phase of LTP in acetylcholinergic synaptic inputs.

As a next step, we tested whether co-application of either NaB or TSA with RG108 was able to prevent the RG108-induced LTP weakening in acetylcholinergic synapses (Figure 3B, groups LTP+RG108+NaB, n = 8, LTP+RG108+TSA, n = 8). Test stimulation before applying the potentiation protocol led to a gradual slight decrease in EPSP amplitude in LTP+RG108+NaB and LTP+RG108+TSA similar to the groups described earlier (control, control+RG108, LTP, LTP+RG108); no significant differences were observed between those groups at the 0 min time point (Figure 3B,D). Five tetanic stimulations combined with 5-HT applications caused a significant increase in the EPSP amplitude: the LTP of synaptic responses was observed in LTP+RG108+NaB and LTP+RG108+TSA groups (Figure 3B). According to the data obtained, the EPSP amplitudes in LTP+RG108+NaB were significantly higher than those in LTP+RG108 for the time window 80–250 min (Figure 3B,C,E,F), while at the early stage of potentiation, differences in amplitudes were found only at time points 20 and 40 min. However, despite the preservation of EPSP amplitudes in LTP+RG108+NaB at potentiated levels throughout the experiment, they were significantly different from those in the LTP group (Figure 3B). Thus, at the time point 250 min, the EPSP amplitudes in the LTP+RG108+NaB group were 98.0 ± 5.3%. Notably, the EPSP amplitudes in the LTP+RG108+TSA were not affected by RG108 administration: the EPSPs in LTP+RG108+TSA did not differ from those in the LTP group throughout the recording (Figure 3B–F, Table 2). Thus, co-application of NaB or TSA with RG108 kept the EPSPs stably elevated. Therefore, it is possible to speculate that HDACis (NaB or TSA) can reverse the RG108-induced LTP deficits in acetylcholinergic synaptic inputs.

## 4. Discussion

We have examined the involvement of the DNMT activity in LTP in withdrawal interneurons. RG108 applications did not significantly affect the EPSP amplitudes in the control and control+RG108 groups when synaptic inputs with different transmitters were activated via second cutaneal or intestinal nerves (Figure 2 and Figure 3, control and control+RG108 groups). This suggested that the DNMT inhibitor used did not affect the synaptic potentials by itself. Similar results were obtained in the study of the effect of DNMTis on basic synaptic transmission in the acute slices of the hippocampus [3]. Next, we examined the effects of RG108 on LTP induction and maintenance. Following the induction protocol using the glutamatergic afferents, the potentiation of synaptic potentials in the LTP+RG108 group was induced and maintained within 60 min after induction, but after the 120 min time point, the synaptic potentials gradually decreased to the non-potentiated EPSP amplitudes (Figure 2, LTP+RG108 vs. LTP, LTP+RG108 vs. control). Thus, the data obtained showed the dependency of maintenance of LTP induced by the second cutaneal nerve tetanizations on DNA methylation. Notably, the RG108 did not completely prevent the LTP, affecting only the late LTP (maintenance), while the early LTP remained untouched. Moreover, the results obtained could not be attributed to a simple decrease in EPSP amplitude by RG108 administration as we did not observe the EPSPs amplitudes changes in recordings under RG108 (Figure 2 and Figure 3, control+RG108 group).

Regarding the induction and maintenance of LTP induced by intestinal nerve tetanizations (putative acetylcholinergic synaptic inputs), we demonstrated that RG108 did not prevent the induction of acetylcholinergic LTP but significantly disrupted its early and late phases. It appeared that RG108 significantly downregulated acetylcholine-mediated EPSPs starting from the time point of 70 min (Figure 3, LTP+RG108 vs. LTP). After the 120 min time point, the EPSP amplitudes in the LTP+RG108 and control groups were not significantly different from each other (Figure 3, LTP+RG108 vs. control). In summary, DNMT inhibition had a profound deteriorating effect on facilitated EPSPs evoked both by glutamatergic and cholinergic afferents. Thus, we can speculate that DNA methylation is required for the development and maintenance of LTP in synaptic inputs of withdrawal interneurons in terrestrial snails.

These data are consistent with previous studies showing that mechanisms of synaptic plasticity are dependent on dynamic changes in DNA methylation/demethylation [45]. In this study, inhibition of DNMT blocked LTP, which suggests that DNMT activity is necessary to induce the necessary level of DNA methylation for the formation and maintenance of long-term synaptic plasticity.

As synaptic plasticity is the main mechanism by which memory and plasticity are realized, it is important to note the studies on the role of DNA methylation in the formation of long-term memory [8,46,47,48,49]. Recently, the effects of inhibition of DNMT on memory maintenance in *Helix* have been characterized. Similarly to the results obtained in this study, RG108 impaired long-term context and cued memory maintenance in *Helix* [14,34]. The emerging picture from these data indirectly suggests an intimate link between synaptic plasticity, memory and DNMT activity in terrestrial snails.

The results described in this paper constitute the first account of the role of DNMT in synaptic plasticity in terrestrial gastropod snails. In vertebrates, the DNMT involvement in LTP has been documented previously. It was demonstrated that the activity of DNMT plays an important role in successful LTP formation. Genetically reduced DNMT levels [12,17] or pharmacological DNMT inhibition [3,50] led to marked deficits in LTP. In addition, it turned out that DNA methylation is also an important regulator of memory-related processes in vertebrates. Pharmacological [10,11,16,31,51,52,53] and genetic studies [12,15,17,54,55,56] have established a relationship between DNMT activity and memory formation: an impairment in DNMT activity abolished memory formation [13]. Taken together, these studies show that DNA methylation is a necessary component of the formation of long-term memory and long-term synaptic plasticity both in invertebrates and vertebrates. It should be added that in our work, the effect of RG108 on DNA methylation has not been directly studied using molecular approaches. However, there are multiple proofs in the literature that the inhibition of DNMT by RG108 leads to DNA demethylation [57,58,59,60]. Based on these results, we assume that the effects of the DNMT inhibitor RG108 found in our work are associated with changes in DNA methylation.

However, we can only speculate about signaling molecules upstream of DNMT. In our experiments, the LTP was induced by repeated exposure to 5-HT with the stimulation of afferents, which corresponds to long-term memory formation through 5-HT release from modulatory serotonergic neurons of pedal ganglia [33]. Further, 5-HT might bind to different G protein-coupled receptors [61]. Those receptors might activate diverse molecular signaling pathways. The downstream signaling pathway after receptors’ activation is extremely vague. However, it is well established that one of the 5-HT targets is protein kinase C (PKC) activation [62,63,64,65,66]. PKC is considered to be one of the most important enzymes in synaptic plasticity [62,67,68]. According to the literature, different PKC isoforms might require different activators (for example, DAG and/or an increase in the intracellular Ca^2+^ [69,70,71]) and lead to different downstream activities. Levenson with colleagues demonstrated that the PKC signaling pathway regulated the expression of DNMT genes [3]. To sum up, 5-HT through G protein-coupled receptors might lead to an increase in the intracellular Ca^2+^ via activation of IP3 receptors and RyRs [72] and DAG synthesis, PKC activation and changes in DNMT gene expression. Alterations of DNMT genes’ expression in turn may cause changes in the methylation status of various plasticity-related genes [9,10,51,73]. The described model is hypothetical and does not exclude the contribution of other possible signaling pathways (Figure 4).

In our study, we also showed that LTP deficiency caused by the DNMTi RG108 was eliminated by a pharmacologically increased level of histone acetylation under NaB or TSA administration (Figure 2 and Figure 3, groups LTP+RG108+NaB, LTP+RG108+TSA). The ability of HDACis NaB and TSA to increase the level of histone acetylation has been shown previously in invertebrates [26] and vertebrates [24,74]. These data are consistent with a number of studies that demonstrated that an increased level of histone acetylation improved long-term synaptic plasticity [25,31,75,76] and reversed its deficiency [29,30]. It should be noted that the results described here are also consistent with our previous results [30,34,35,36]. Using behavioral approaches, we demonstrated that HDACis NaB and TSA might act as cognitive enhancers for weak or impaired memories.

Monsey et al. [16] and Miller et al. [31] also observed that DNMT inhibition resulted in significant LTP impairment, while co-application of DNMTi with HDACi reverses the deficit in LTP. Besides the fact that the data obtained emphasize the role of DNA methylation and histone acetylation as important epigenetic mechanisms for regulating synaptic plasticity in withdrawal interneurons, they also support the idea that DNA methylation and histone acetylation may affect each other to regulate the LTP [3]. Speaking of this, it should be noted that according to the results of several studies, DNMT inhibition may prevent histone acetylation [16,31,77,78]. The possible model through which HDAC might affect synaptic plasticity in withdrawal interneurons is described elsewhere (see [30]).

Our study has some limitations. Firstly, we used only RG108 for DNA methyltransferases inhibition and two structurally distinct histone deacetylases inhibitors (sodium butyrate and trichostatin A). The work would undoubtedly benefit from using more inhibitors. However, since the chosen electrophysiological technique is very labor-intensive, we decided to focus on those three blockers. Taking into account quite a few published studies in this field in gastropod snails, we completely focused on electrophysiological research, and carried out a large amount of experimental work, using all the necessary controls. Indeed, the work would undoubtedly benefit from molecular data, which we plan to obtain in future studies. This was not within the scope of the current study.

## 5. Conclusions

Despite the fact that terrestrial snail *Helix lucorum* is a representative of the largest class of gastropods, at the moment, the question of the role of DNA methylation in its synaptic plasticity is only beginning to be investigated. In this work, it was shown for the first time that the blockade of DNMT disrupts the maintenance of LTP in withdrawal interneurons caused by afferent stimulation (cutaneal or intestinal nerves) and 5-HT applications. This fact allows us to consider the activity of DNMT, and, consequently, of DNA methylation, as an important regulatory mechanism and a necessary condition for the development and maintenance of long-lasting synaptic changes in terrestrial snails.

In addition, the experiments conducted suggest that LTP deficits, impaired by DNMT inhibition, can be reversed by blocking histone deacetylation. These results confirm the idea of the interaction of DNA methylation and histone acetylation in the epigenetic regulation of synaptic plasticity. Moreover, the data obtained in combination with the present observations demonstrate that epigenetic modulation of DNA methylation and histone acetylation are observed both in vertebrates and invertebrates (*Helix lucorum*) and as an evolutionarily conservative phenomena reflect the main characteristics of long-term synaptic plasticity.

## Figures and Tables

**Figure 1 cells-13-01850-f001:**
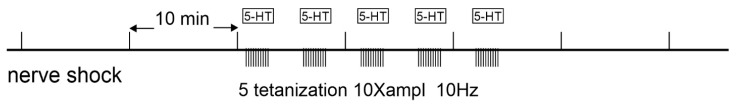
Schematic representations of protocol. 5-HT—serotonin.

**Figure 2 cells-13-01850-f002:**
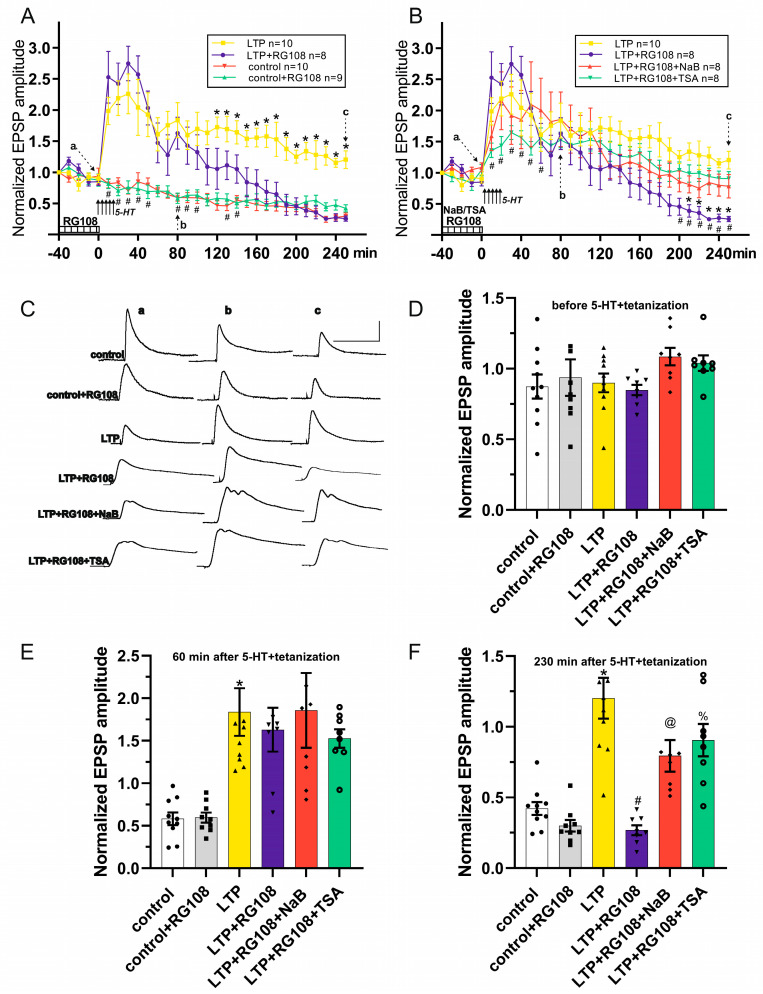
The effects of RG108, sodium butyrate (NaB) and trichostatin A (TSA) on EPSP amplitudes in control, under LTP induction and LTP maintenance in withdrawal interneurons with test stimulations of the second cutaneal nerve. (**A**) Effects of the bath applications of RG108 on the EPSPs: a time course plot of pooled data for the EPSPs. RG108 reduces LTP duration. * denotes *p* < 0.05 LTP+RG108 vs. LTP; # denotes *p* < 0.05 LTP+RG108 v vs. control. (**B**) Effect of the combined bath application of RG108 with either NaB or TSA on the EPSPs: a time course plot of pooled data for the EPSPs. Histone deacetylase inhibitors (NaB or TSA) prevented the weakening of potentiation. * denotes *p* < 0.05 LTP+RG108+NaB vs. LTP+RG108; # denotes *p* < 0.05 LTP+RG108+TSA vs. LTP+RG108. The data are presented as mean  ±  standard error of the mean. The duration of drugs’ presence is shown as a striped bar at the bottom in (**A**,**B**). Arrows indicate the time of tetanization+5-HT applications in (**A**,**B**). (**C**) Example traces of the EPSPs (a, b and c from plots (**A**,**B**)) in control, control+RG108, LTP, LTP+RG108, LTP+RG108+NaB and LTP+RG108+TSA groups. Scale bars:5 mV, 500 ms. (**D**) Comparison of the effects of RG108, NaB and TSA before the tetanization+5-HT. No differences were observed between EPSP amplitudes in groups before 5-HT+tetanization (time point 0 min) (**E**) Comparison of the effects of RG108, NaB and TSA on LTP induction: EPSP amplitudes 60 min post-5-HT+tetanization (time point of 80 min). EPSP amplitudes of LTP group were significantly higher than those in control; * denotes *p* < 0.05 LTP vs. control. Moreover, there were no significant differences between the groups with LTP induction. (**F**) Comparison of the effects of RG108, NaB and TSA on LTP maintenance: EPSP amplitudes 230 min post-5-HT+tetanization (time point 250 min). Application of RG108 caused a significant decrease in EPSP amplitude, while co-administration of RG108+NaB or RG108+TSA was shown to enhance LTP. * denotes *p* < 0.05 LTP vs. control; # denotes *p* < 0.05 LTP vs. LTP+RG108; @ denotes *p* < 0.05 LTP+RG108 vs. LTP+RG108+NaB; % denotes *p* < 0.05 LTP+RG108 vs. LTP+RG108+TSA.

**Figure 3 cells-13-01850-f003:**
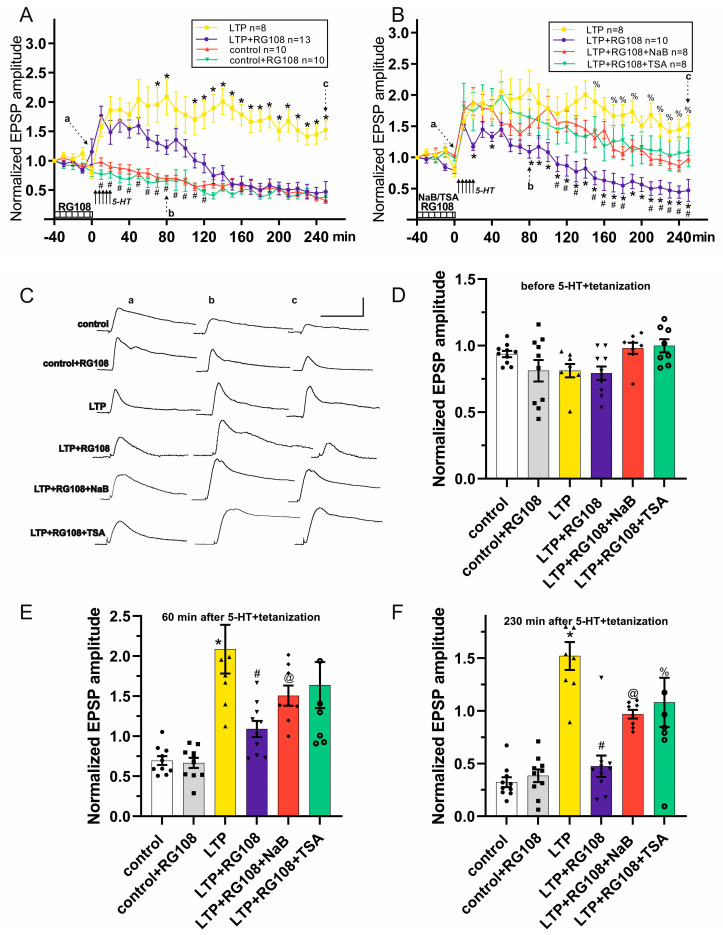
The effects of RG108, NaB and TSA on EPSP amplitudes in control, under LTP induction and LTP maintenance in withdrawal interneurons with test stimulation of intestinal nerve. (**A**) Effects of the bath application of RG108 on the EPSPs: a time course plot of averaged data for the EPSPs. RG108 reduces the LTP maintenance. * denotes *p* < 0.05 in groups LTP+RG108 vs. LTP; # denotes *p* < 0.05 LTP+RG108 vs. control. (**B**) Effect of the combined bath application of RG108 with either NaB or TSA on the EPSPs: a time course plot of averaged data for the EPSPs. Histone deacetylase inhibitors (NaB or TSA) prevented the weakening of potentiation. * denotes *p* < 0.05 in groups LTP+RG108+NaB vs. LTP+RG108; # denotes *p* < 0.05 LTP+RG108+TSA vs. LTP+RG108. % denotes *p* < 0.05 LTP+RG108+NaB vs. LTP. The data are presented as mean ± SEM. The duration of drug presence is shown as a striped bar at the bottom in (**A**,**B**). Arrows indicate the time of tetanization+5-HT in (**A**,**B**). (**C**) Examples of the EPSPs (a, b and c from plots (**A**,**B**)) in control, control+RG108, LTP, LTP+RG108, LTP+RG108+NaB and LTP+RG108+TSA groups. Scale bars:5 mV, 500 ms. (**D**) Comparison of the effects of RG108, NaB and TSA before the tetanization+5-HT. No differences were observed between EPSP amplitudes in groups before the 5-HT+tetanizations (time point 0 min) (**E**) Comparison of the effects of RG108, NaB and TSA on LTP induction: EPSP amplitudes 60 min post-5-HT+tetanization (time point 80 min). EPSP amplitudes of LTP group were significantly higher than those in control and LTP+RG108 groups; * denotes *p* < 0.05 LTP vs. control; # denotes *p* < 0.05 LTP vs. LTP+RG108; @ denotes *p* < 0.05 LTP+RG108+NaB vs. LTP+RG108. (**F**) Comparison of the effect of RG108, NaB and TSA on LTP maintenance: EPSP amplitudes 230 min post-5-HT+tetanization (time point 250 min). Application of RG108 caused a significant decrease in EPSP amplitude while co-administration of RG108+NaB or RG108+TSA was shown to enhance LTP. * denotes *p* < 0.05 LTP vs. control; # denotes *p* < 0.05 LTP vs. LTP+RG108; @ denotes *p* < 0.05 LTP+RG108 vs. LTP+RG108+NaB; % denotes *p* < 0.05 LTP+RG108 vs. LTP+RG108+TSA.

**Figure 4 cells-13-01850-f004:**
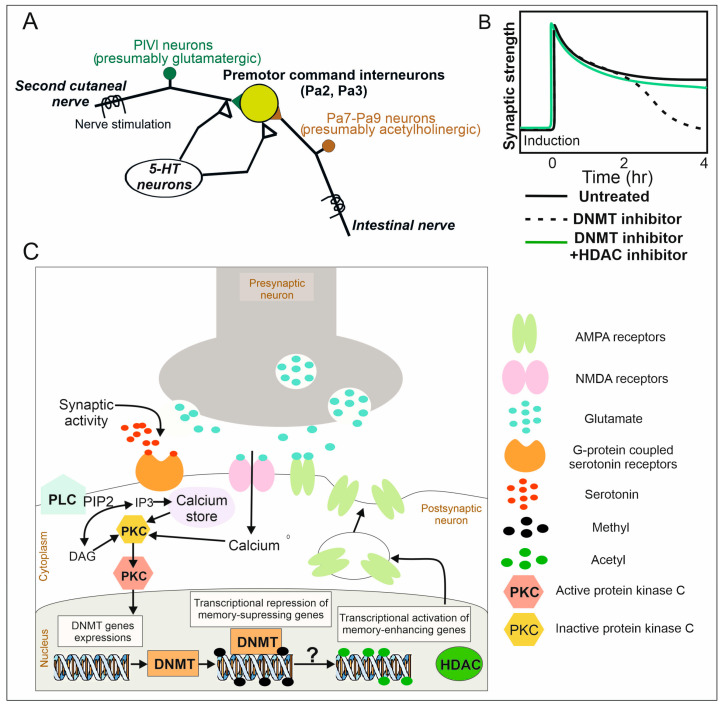
Epigenetic regulation of LTP in the withdrawal behavior of interneurons in terrestrial snails. (**A**). The neural network underlying LTP formation. (**B**). Inhibition of DNMT activity inhibits late LTP without affecting induction and early LTP, while the combined use of HDACi with DNMTi prevents potentiation attenuation. (**C**). Proposed model of epigenetic regulation of LTP in premotor interneurons with the example of glutamatergic presynaptic input. The induction and stabilization of LTP require NMDA and 5-HT receptor activation. This leads to the burst of intracellular calcium, synthesis of inositol 1,4,5-trisphosphate (IP3, mobilizing calcium from calcium stores) and diacylglycerol (DAG) from phosphatidylinositol 4,5-bisphosphate (PIP2) (phospholipase C (PLC) catalyses the hydrolysis of PIP2). DAG and/or an increase in the intracellular calcium act as activators of protein kinase C (PKC). The latter regulates the expression of DNMT genes. Alterations of DNMT genes’ expression in turn may cause changes in the methylation status of various plasticity-related genes. Moreover, DNA methylation somehow (“?” on the picture) affects the histone acetylation level. Successful activation of these signaling pathways leads to an increase in the number of AMPA receptors and, consequently, the synaptic strength.

**Table 1 cells-13-01850-t001:** Summary table showing the statistical significance of differences between all groups for second cutaneal nerve (*p* < 0.05 is considered significant).

	Control	Control	Control	Control	Control	Control+RG	Control+RG	Control+RG	Control+RG	LTP	LTP	LTP	LTP+RG	LTP+RG	LTP+RG+NaB
	vs.	vs.	vs.	vs.	vs.	vs.	vs.	vs.	vs.	vs.	vs.	vs.	vs.	vs.	vs.
	Contr+RG	LTP	LTP+RG	LTP+RG+NaB	LTP+RG+TSA	LTP	LTP+RG	LTP+RG+NaB	LTP+RG+TSA	LTP+RG	LTP+RG+NaB	LTP+RG+TSA	LTP+RG+NaB	LTP+RG+TSA	LTP+RG+TSA
−40	>0.999	>0.999	>0.999	>0.999	>0.999	>0.999	>0.999	>0.999	>0.999	>0.999	>0.999	>0.999	>0.999	>0.999	>0.999
−30	0.094	0.133	0.006	0.011	0.541	0.278	0.370	0.370	0.200	0.016	0.237	0.460	0.279	0.015	0.130
−20	>0.999	0.252	0.481	0.963	0.681	0.299	0.606	0.606	0.837	0.189	0.281	0.710	0.721	0.397	0.779
−10	0.666	0.720	0.423	0.031	0.681	0.549	>0.999	>0.999	>0.999	0.460	0.161	0.536	0.021	0.867	0.165
0	0.931	0.842	>0.999	0.174	0.364	0.720	0.898	0.898	0.298	0.513	0.109	0.440	0.048	0.056	0.876
10	>0.999	0.000	0.000	0.001	0.005	0.000	0.000	0.000	0.003	0.313	0.315	0.109	0.073	0.014	>0.999
20	0.258	0.000	0.000	0.002	0.008	0.000	0.000	0.000	0.002	0.573	0.859	0.088	0.524	0.014	0.268
30	0.546	0.000	0.000	0.000	0.000	0.000	0.000	0.000	0.000	0.281	0.445	0.232	0.228	0.003	>0.999
40	0.743	0.029	0.000	0.059	0.003	0.023	0.000	0.000	0.000	0.536	0.731	0.336	0.228	0.010	0.950
50	0.340	0.004	0.000	0.002	0.002	0.004	0.000	0.000	0.001	0.696	0.813	0.408	0.536	0.050	0.613
60	0.931	0.010	0.059	0.000	0.000	0.008	0.059	0.059	0.000	0.633	0.696	0.965	0.505	0.574	0.721
70	0.666	0.000	0.071	0.000	0.000	0.000	0.142	0.142	0.000	0.070	0.696	0.408	0.336	0.281	0.878
80	>0.999	0.000	0.001	0.000	0.000	0.000	0.001	0.001	0.000	0.965	0.897	0.962	0.721	>0.999	0.955
90	0.730	0.000	0.008	0.001	0.000	0.000	0.015	0.015	0.000	0.460	0.762	0.965	0.645	0.574	0.878
100	0.815	0.000	0.038	0.001	0.003	0.000	0.036	0.036	0.002	0.237	0.515	>0.999	0.645	0.382	0.798
110	0.796	0.000	0.114	0.002	0.001	0.000	0.114	0.114	0.002	0.083	0.573	0.813	0.130	0.189	0.694
120	0.436	0.000	0.174	0.001	0.001	0.000	0.408	0.408	0.005	0.043	0.122	0.601	0.336	0.165	0.613
130	0.489	0.000	0.008	0.008	0.002	0.000	0.021	0.021	0.003	0.043	0.055	0.270	>0.999	0.232	0.397
140	0.546	0.000	0.036	0.015	0.008	0.000	0.021	0.021	0.005	0.034	0.068	0.193	0.645	0.281	0.463
150	0.931	0.000	0.277	0.006	0.005	0.000	0.673	0.673	0.008	0.016	0.101	0.230	0.161	0.281	0.463
160	0.489	0.000	0.321	0.015	0.002	0.000	0.743	0.743	0.006	0.016	0.101	0.315	0.161	0.105	0.382
170	0.931	0.000	0.370	0.008	0.008	0.000	0.743	0.743	0.008	0.009	0.034	0.193	0.161	0.232	0.536
180	0.730	0.000	0.536	0.015	0.008	0.000	>0.999	>0.999	0.011	0.025	0.055	0.173	0.232	0.232	0.505
190	0.258	0.000	0.470	0.027	0.002	0.000	0.408	0.408	0.015	0.003	0.173	0.173	0.072	0.054	0.959
200	0.796	0.000	0.864	0.093	0.002	0.000	0.689	0.689	0.002	0.001	0.122	0.237	0.081	0.008	0.442
210	0.387	0.000	0.681	0.042	0.002	0.000	0.299	0.299	0.015	0.000	0.025	0.101	0.026	0.004	0.463
220	0.077	0.000	0.536	0.050	0.000	0.003	0.091	0.091	0.015	0.001	0.118	0.408	0.051	0.002	0.345
230	0.054	0.000	0.606	0.003	0.000	0.000	0.031	0.031	0.008	0.000	0.118	0.237	0.005	0.000	0.755
240	0.077	0.000	0.776	0.005	0.000	0.000	0.145	0.145	0.011	0.000	0.093	0.083	0.015	0.001	0.662
250	0.189	0.000	0.534	0.010	0.001	0.000	0.029	0.029	0.005	0.000	0.099	0.203	0.004	0.001	0.354

**Table 2 cells-13-01850-t002:** Summary table showing the statistical significance of differences between all groups for intestinal nerve (*p* < 0.05 is considered significant).

	Control	Control	Control	Control	Control	Control+RG	Control+RG	Control+RG	Control+RG	LTP	LTP	LTP	LTP+RG	LTP+RG	LTP+RG+NaB
	vs.	vs.	vs.	vs.	vs.	vs.	vs.	vs.	vs.	vs.	vs.	vs.	vs.	vs.	vs.
	Contr+RG	LTP	LTP+RG	LTP+RG+NaB	LTP+RG+TSA	LTP	LTP+RG	LTP+RG+NaB	LTP+RG+TSA	LTP+RG	LTP+RG+NaB	LTP+RG+TSA	LTP+RG+NaB	LTP+RG+TSA	LTP+RG+TSA
−40	>0.999	0.444	>0.999	>0.999	>0.999	0.444	>0.999	>0.999	>0.999	>0.999	>0.999	>0.999	>0.999	>0.999	>0.999
−30	0.143	>0.999	0.077	0.965	0.122	0.083	0.738	0.122	0.203	0.121	0.959	0.279	0.064	0.414	0.050
−20	0.549	0.541	0.702	0.888	>0.999	0.460	0.722	0.274	0.897	0.678	0.959	0.328	0.473	0.734	0.721
−10	0.739	0.093	0.483	0.016	0.042	0.147	0.232	0.021	0.118	0.022	0.852	0.937	0.001	0.007	0.755
0	0.497	0.199	0.460	0.027	0.529	0.733	0.512	0.055	0.263	0.605	0.008	0.257	0.085	0.272	0.491
10	0.105	0.003	0.000	0.000	0.014	0.000	0.000	0.000	0.002	0.547	0.463	0.955	0.643	0.536	0.710
20	0.661	0.001	0.004	0.000	0.008	0.000	0.001	0.000	0.001	0.121	0.779	0.505	0.037	0.804	0.397
30	0.143	0.000	0.000	0.001	0.004	0.000	0.000	0.000	0.000	0.270	0.867	0.798	0.592	0.792	0.955
40	0.190	0.000	0.000	0.000	0.002	0.000	0.000	0.000	0.000	0.140	0.694	0.878	0.157	0.547	0.867
50	0.971	0.000	0.000	0.000	0.000	0.000	0.000	0.001	0.000	0.500	0.281	0.878	0.938	0.185	0.232
60	0.165	0.000	0.001	0.001	0.000	0.000	0.000	0.000	0.000	0.161	0.382	0.721	0.595	0.268	0.721
70	0.481	0.000	0.000	0.001	0.000	0.000	0.000	0.001	0.000	0.045	0.130	0.442	0.750	0.238	0.574
80	0.912	0.000	0.001	0.000	0.000	0.000	0.002	0.000	0.000	0.008	0.161	0.195	0.140	0.336	0.959
90	0.105	0.000	0.003	0.000	0.001	0.001	0.003	0.001	0.002	0.076	0.645	0.328	0.140	0.547	0.645
100	0.243	0.001	0.006	0.000	0.001	0.002	0.008	0.001	0.001	0.140	0.959	0.574	0.013	0.456	0.328
110	0.842	0.000	0.011	0.000	0.000	0.000	0.006	0.000	0.000	0.020	>0.999	0.328	0.019	0.089	0.397
120	0.218	0.000	0.088	0.000	0.001	0.000	0.008	0.000	0.000	0.003	0.645	0.234	0.013	0.053	0.505
130	0.035	0.000	0.648	0.000	0.003	0.000	0.026	0.000	0.000	0.000	0.336	0.065	0.002	0.020	0.281
140	0.971	0.000	0.313	0.000	0.006	0.000	0.376	0.000	0.001	0.000	0.065	0.065	0.005	0.053	0.328
150	0.043	0.000	0.872	0.001	0.002	0.000	0.254	0.000	0.000	0.000	0.038	0.161	0.000	0.003	0.959
160	0.631	0.000	0.771	0.000	0.012	0.000	0.872	0.000	0.003	0.000	0.195	0.065	0.001	0.010	0.195
170	0.684	0.000	0.418	0.001	0.016	0.000	0.628	0.000	0.003	0.000	0.038	0.234	0.002	0.007	0.959
180	0.247	0.000	0.674	0.001	0.009	0.000	0.628	0.000	0.001	0.000	0.007	0.161	0.003	0.004	0.721
190	0.280	0.000	0.918	0.000	0.055	0.000	0.387	0.000	0.016	0.000	0.038	0.065	0.001	0.075	0.645
200	0.529	0.000	0.973	0.000	0.068	0.000	0.512	0.000	0.055	0.000	0.065	0.279	0.002	0.051	0.959
210	0.631	0.000	0.796	0.001	0.016	0.000	0.529	0.000	0.003	0.000	0.006	0.072	0.004	0.027	0.959
220	0.853	0.000	0.739	0.001	0.021	0.000	0.529	0.000	0.009	0.001	0.028	0.130	0.003	0.027	0.959
230	0.579	0.000	0.796	0.001	0.016	0.000	0.481	0.001	0.016	0.001	0.048	0.234	0.006	0.027	>0.999
240	>0.999	0.000	0.720	0.001	0.016	0.000	0.780	0.000	0.006	0.001	0.009	0.232	0.011	0.036	0.959
250	0.278	0.000	0.607	0.003	0.008	0.000	0.875	0.002	0.004	0.008	0.042	0.152	0.014	0.008	0.933

## Data Availability

Data will be provided upon reasonable request.

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
