# Peer review of "DNA Methylation and Histone Acetylation Contribute to the Maintenance of LTP in the Withdrawal Behavior Interneurons in Terrestrial Snails"

_cells, 2024, doi:10.3390/cells13221850_

Round 1
Reviewer 1 Report
Comments and Suggestions for Authors
The article presented by Zuzina et al., investigates the role of epigenetic mechanisms—specifically DNA methylation and histone acetylation—in long-term potentiation (LTP) in terrestrial snail Helix lucorum. The study utilizes electrophysiological techniques to show that DNA methylation is critical for the maintenance of LTP, and that inhibition of DNA methyltransferases (DNMTs) with RG108 impairs LTP. Interestingly, histone deacetylase inhibitors (HDACis) like sodium butyrate (NaB) and trichostatin A (TSA) can rescue the RG108-induced LTP deficits, demonstrating a possible interaction between DNA methylation and histone acetylation in regulating synaptic plasticity.
General comments
I consider this study novelty, because it examines synaptic plasticity in a non-vertebrate model, the terrestrial snail Helix lucorum, which provides a simplified but relevant system for understanding neuroplasticity.
The exploration of DNA methylation and histone acetylation adds valuable insights into the role of epigenetics in LTP, a critical process for memory and learning. The use of electrophysiology to record synaptic events is a strong aspect of the methodology, providing direct functional insights into the role of DNMT inhibition and HDACi co-application in synaptic plasticity.
The inclusion of control groups and the clear experimental design (e.g., RG108 application, NaB/TSA co-application) allow for rigorous comparison across conditions.
Major comments
While the authors hypothesize that RG108 disrupts DNA methylation, there is no direct molecular evidence in this study (e.g., methylation status of DNA). Future experiments could include bisulfite sequencing or methylation-specific PCR to confirm DNA methylation changes following RG108 treatment.
The study uses only one DNMT inhibitor (RG108), which limits the generalizability of the findings. Including additional DNMT inhibitors (e.g., 5-azacytidine) would strengthen the conclusions about the role of DNA methylation in LTP. HDAC inhibitors (e.g., vorinostat) also would provide a more robust understanding of how these pathways interact to regulate LTP.
While the article focuses on histone acetylation, it does not measure histone modifications directly, and . Adding chromatin immunoprecipitation (ChIP) assays could provide a more direct link between histone modifications and LTP, particularly by showing changes in histone acetylation at genes relevant to synaptic plasticity.
While the study focuses on snails, it would benefit from a discussion that ties these findings to vertebrate models. This would help bridge the gap between invertebrate and vertebrate synaptic plasticity and make the study more relevant for broader neuroscience audiences.
As previously mentioned, a bisulfite sequencing experiment to detect methylation changes in specific genes would complement the functional findings and provide mechanistic insights. To confirm the effect of NaB and TSA on histone acetylation at specific loci, performing ChIP-qPCR or ChIP-seq would help demonstrate that HDAC inhibition leads to acetylation at genes related to plasticity and memory.
If feasible, linking the synaptic changes to behavior would provide additional depth to the study. For example, pairing the LTP findings with behavioral conditioning assays could reveal how DNA methylation and histone acetylation affect memory formation in Helix lucorum.
The figure 2 illustrating the experimental design for inducing LTP is clear and essential for understanding the methodology. However, it would be helpful to include a graphical representation of the changes in EPSPs over time for all groups.
In Fig. 2. the time course plots for the different experimental conditions are well presented. However, including error bars for all data points would improve clarity. It would also be helpful to have a summary table showing the statistical significance of differences between all groups.
The figure 3 shows the effects of different treatments on EPSP amplitudes, but in my opinión, the presentation could be enhanced by incorporating individual data points (scatter plots) overlaid on the time-course graph to show data distribution.
Overall, I consider that the study makes significant contributions to our understanding of the role of epigenetics in synaptic plasticity, with an interesting model, However, the addition of molecular assays and the use of additional pharmacological tools would greatly enhance the mechanistic insights and impact of the study.
Author Response
Dear Editors,
Please find our point-to-point responses to the Reviewers comments and a revised version of the paper. In general, we are very thankful to the Reviewers for detailed comments and analysis of our results.
Responses are marked in Red
- Zuzina, D. Kolotova, P. Balaban
Reviewer
Major comments
1.While the authors hypothesize that RG108 disrupts DNA methylation, there is no direct molecular evidence in this study (e.g., methylation status of DNA). Future experiments could include bisulfite sequencing or methylation-specific PCR to confirm DNA methylation changes following RG108 treatment.
Thank you for pointing this out.
Indeed, the effect of RG108 on DNA methylation was not tested in the current work. Bisulfite sequencing or methylation-specific PCR would provide direct molecular evidence. However, such studies, to date, are not possible, since we do not have the genome assembly and annotation for Helix lucorum.
2.The study uses only one DNMT inhibitor (RG108), which limits the generalizability of the findings. Including additional DNMT inhibitors (e.g., 5-azacytidine) would strengthen the conclusions about the role of DNA methylation in LTP. HDAC inhibitors (e.g., vorinostat) also would provide a more robust understanding of how these pathways interact to regulate LTP.
Thank you for pointing this out.
Only RG108 was used in all experiments for DNMT inhibition. The choice of RG108 as DNMT blocker in the current work was due to the fact that RG108 was shown to bind the active site of DNMT and inhibits the enzyme by blocking the catalytic domain without the need for incorporation into DNA. (Schirrmacher E, Beck C, Brueckner B, Schmitges F, Siedlecki P, Bartenstein P, and others. 2006. Synthesis and in vitro evaluation of biotinylated RG108: a high affinity compound for studying binding interactions with human DNA methyltransferases. Bioconjug Chem 17:261–6.). At the same time, to block the DNMT function, the 5-Aza-2′-deoxycytadine is incorporated into DNA and trap DNMTs by covalent binding of the enzyme to DNA. Unlike RG108, the nucleoside inhibitors (5-Aza-2′-deoxycytadine or zebularine) are much more cytotoxic (Creusot F, Acs G, Christman JK. 1982. Inhibition of DNA methyltransferase and induction of Friend erythroleukemia cell differentiation by 5-azacytidine and 5-aza-2′-deoxycytidine. J Biol Chem 257:2041–8.).
Moreover, as we wrote in the article, there is a large amount of literature indicating that RG108 causes DNA demethylation [57–60]. This blocker has been repeatedly used to study the mechanisms of synaptic plasticity, learning and memory in experiments on various groups of both invertebrates and vertebrates, including gastropod snails, where the effect of RG108 was explained precisely by its effect on DNA methylation, and not on any other targets (Pearce, K.; Cai, D.; Roberts, A.C.; Glanzman, D.L. Role of protein synthesis and DNA methylation in the consolidation and maintenance of long-term memory in Aplysia. eLife 2017, 6, e18299. Zuzina, A.B.; Vinarskaya, A.K.; Balaban, P.M. DNA Methylation Inhibition Reversibly Impairs the Long-Term Context Memory Maintenance in Helix. Int. J. Mol. Sci. 2023, 24, 14068. Yang, Q.; Antonov, I.; Castillejos, D.; Nagaraj, A.; Bostwick, C.; Kohn, A.; Moroz, L.L.; Hawkins, R.D. Intermediate-term memory in Aplysia involves neurotrophin signaling, transcription, and DNA methylation. Learn. Mem. Cold Spring Harb. N 2018, 25, 620–628. Maddox SA, Watts CS, Schafe GE. DNA methyltransferase activity is required for memory-related neural plasticity in the lateral amygdala. Neurobiol Learn Mem. 2014 Jan;107:93-100. Maddox, S.A.; Schafe, G.E. Epigenetic alterations in the lateral amygdala are required for reconsolidation of a Pavlovian fear memory. Learn. Mem. Cold Spring Harb. N 2011, 18, 579–593).
As for histone deacetylase inhibitors, we used two structurally distinct inhibitors sodium butyrate and trichostatin A that inhibit different classes of HDAC enzymes according to the literature.
We added the section “Limitations” where this issue is indicated. Changes can be found on page 18, line 471.
Limitations of the study: Our study have some limitations. Firstly, we used only RG108 for DNA methyltransferases inhibition and two structurally distinct HDAC inhibitors histone deacetylases inhibitors (sodium butyrate and trichostatin A). The work would undoubtedly benefit from using more inhibitors. However, since the chosen electrophysiological technique is very labor-intensive, we decided to focus on those three blockers. Taking into account quite a few published studies in this field in gastropod snail, we have completely focused on electrophysiological research, and carried out a large amount of experimental work, using all the necessary controls. Of course, the work would undoubtedly benefit from molecular data, which we plan to obtain in future studies. This was not within the scope of the current study.
3.While the article focuses on histone acetylation, it does not measure histone modifications directly, and . Adding chromatin immunoprecipitation (ChIP) assays could provide a more direct link between histone modifications and LTP, particularly by showing changes in histone acetylation at genes relevant to synaptic plasticity.
Thank you for pointing this out.
Adding chromatin immunoprecipitation assays is currently impossible because we do not have the genome assembly and annotation for Helix lucorum. We added the section “Limitations” where this issue is indicated. Changes can be found on page 18, line 471.
While the article doesn’t provide direct measures of histone acetylation, it is based on a large body of literature evidence demonstrating that histone acetylation levels change during long-term plasticity. A significant increase of histone acetylation was shown both in vitro and in vivo ex-periments following the induction of long-term plasticity changes. Acetylation of his-tone H3 was increased in vitro in the hippocampus and in vivo, the contextual fear conditioning was accompanied by similar increases in histone acetylation within the hippocampus 1 hour after training [24]. Treatment with 5-HT increased the acetyla-tion of both histone H3 and H4 at the C/EBP promoter 90 minutes after 5-HT treatment [25]. Danilova and Grinkevich [26] observed a significant increase of histone H3 acet-ylation in the subesophageal complex of ganglia in gastropod mollusk Helix 1 hour af-ter learning. Collectively, a relatively large body of evidence suggests that histone acetylation levels change in the first few hours following learning. Changes in text can be found on page 2, line 61.
4.While the study focuses on snails, it would benefit from a discussion that ties these findings to vertebrate models. This would help bridge the gap between invertebrate and vertebrate synaptic plasticity and make the study more relevant for broader neuroscience audiences.
Thank you for pointing this out.
Although the results obtained is the first account of a role for DNMT in synaptic plasticity in terrestrial gastropod snails, in vertebrates the DNMT involvement for LTP has been documented previously. It was demonstrated that the activity of DNMT is required for the successful LTP formation. Genetically reduced DNMT levels [12,17] or pharmacological DNMT inhibition [3,50] led to marked deficits in LTP. In addition, it turned out that DNA methylation is also an important regulator of memory-related processes in vertebrates. Pharmacological [10,11,16,31,51–53] and genetic studies [12,15,17,54–56] have established a relation-ship between the DNMT activity and memory formation: an impairment in DNMT ac-tivity abolished memory formation [13]. Taken together, these studies show that DNA methylation is a necessary component of the formation of long-term memory and long-term synaptic plasticity both in invertebrates and vertebrates. Changes can be found on page 16, line 393.
Moreover, the data obtained in combination with the present observations demon-strate that epigenetic modulation of DNA methylation and histone acetylation are ob-served both in vertebrates and invertebrates (Helix lucorum) and as an evolutionarily conservative phenomena reflect the main characteristics of long-term synaptic plas-ticity. Changes can be found on page 18, line 457.
5.As previously mentioned, a bisulfite sequencing experiment to detect methylation changes in specific genes would complement the functional findings and provide mechanistic insights. To confirm the effect of NaB and TSA on histone acetylation at specific loci, performing ChIP-qPCR or ChIP-seq would help demonstrate that HDAC inhibition leads to acetylation at genes related to plasticity and memory.
Thank you for pointing this out.
Since we are currently unable to conduct molecular studies, we rely on the available publications to evaluate the effects of HDACis NaB and TSA.
The ability of HDACis NaB and TSA to increase the level of histone acetylation has been shown previously in invertebrates [26] and vertebrates [24,74]. Changes can be found on page 17, line 428.
6.If feasible, linking the synaptic changes to behavior would provide additional depth to the study. For example, pairing the LTP findings with behavioral conditioning assays could reveal how DNA methylation and histone acetylation affect memory formation in Helix lucorum.
Thank you for pointing this out.
As synaptic plasticity is the main mechanism by which memory and plasticity are realized, it’s important to concern the studies on the role of DNA methylation in the formation of long-term memory [8,46–49]. Recently, the effects of inhibition of DNMT on memory maintenance in Helix have been characterized. Similarly to the results ob-tained in this study, RG108 impaired long-term context and cued memory mainte-nance in Helix [14,34]. The emerging picture from these data indirectly suggests an in-timate link between synaptic plasticity, memory and DNMT activity in terrestrial snails. Changes can be found on page 16, line 385.
It should be noted that the results described here are also consistent with our previous results [30,34–36]. Using behavioral approaches, we demonstrated that HDACis NaB and TSA might act as cognitive enhancers for weak or impaired memories. Changes can be found on page 17, line 432.
7.The figure 2 illustrating the experimental design for inducing LTP is clear and essential for understanding the methodology. However, it would be helpful to include a graphical representation of the changes in EPSPs over time for all groups.
Figure 2, panel C (as well as figure 3, panel C) includes a graphical representation of the changes in EPSPs over time for all groups. Each three traces shows EPSPs recorded from one of the group (groups: Control, Control+RG108, LTP, LTP+RG108, LTP+RG108+NaB, LTP+RG108+TSA) before LTP induction (mark “a” – at time point 0 minute) and after LTP induction (marks “b” – at time point 80 minute and “c” – at time point 250 minute).
8.In Fig. 2. the time course plots for the different experimental conditions are well presented. However, including error bars for all data points would improve clarity. It would also be helpful to have a summary table showing the statistical significance of differences between all groups.
Error bars were already included for all data. Perhaps, at some time points, these errors are very small and at this scale are poorly visible (such as Figure 2B, time point 230 or 250).
Summary tables (Table 1, 2) showing the statistical significance of differences between all groups were added. Changes can be found on page 9-10 (Table 1) and page 14-15 (Table 2).
9.The figure 3 shows the effects of different treatments on EPSP amplitudes, but in my opinión, the presentation could be enhanced by incorporating individual data points (scatter plots) overlaid on the time-course graph to show data distribution.
Figures 2 and 3 (Panels D-F) were changed. Changes can be found on page 6 and page 12.
10.Overall, I consider that the study makes significant contributions to our understanding of the role of epigenetics in synaptic plasticity, with an interesting model, However, the addition of molecular assays and the use of additional pharmacological tools would greatly enhance the mechanistic insights and impact of the study.
As we wrote above, the lack of the genome assembly and annotation of Helix lucorum does not allow us to conduct molecular research at the moment, which we have indicated in a specially dedicated section Limitations. In addition, since the role of DNA methylation in LTP in Helix had not been studied before, we focused on electrophysiological type of research. Additional molecular research is necessary in the future. As electrophysiological studies are quite laborious and time consuming, we chose only one DNMTi RG108 but two differed in structure and targets HDACis NaB and TSA. Undoubtedly, the use of additional pharmacological tools would greatly enhance the mechanistic insights and impact of the study.

Reviewer 2 Report
Comments and Suggestions for Authors
Dr. Zuzina, Dr. Balanban and their colleagues aim to understand the roles of two epigenetic modifications, DNA methylation and histone acetylation, in isolated central nervous system of terrestrial snails in vitro. They found that DNMT inhibitor RG108 suppressed the long-lasting potentiation caused by 5-HT application, while co-application of HDACis with RG108 alleviated the weakening of potentiation by dysregulated DNA methylation. Paper is well written and data are clear. Here are some minor revisions needed for publication.
1. Line 60: It was mentioned that serotonylation and acetylation contribute to LTP formation and maintenance. It remains
2. Intro: More discussions/intro should be added here: for example, what is the changes of DNA methylation and histone acetylation upon 5-HT application and during the initial phase of LTP, middle phase and late phase of LTP? Understanding the level changes of these two modifications at different phases of LTP is important. Please refer to relative papers if the DNA methylation dynamics were studied and known before.
3. Before showing results, please explain why you chose second cutaneal nerve and intestinal nerve.
3. Generally at least two different inhibitors should be used to rule out the off target issue. For DNMTi here, only RG108 is used in all experiments. Also, line 346 you mentioned that the effect of RG108 on DNA methylation has not been directly studied. It is a big concern. Please explain why only this one inhibitor was chosen before showing the result.
4. For Figure 2A and 1B panel: add 5-HT to the panel to illustrate when (30min) it was applied.
5. Figure 2: it is interesting to see HDACi applications decreased the amplitude in the very late stage: 210-250 min. A key question is: when do you see the increased histone acetylation after applying these inhibitors? It will be very helpful to have a western blot using pan Acetylation Ab.
6. In the future studies, it will be interesting to profile the misregulated genes in these nerves upon application of DNMTi and/or HDACi.
Round 2
Reviewer 1 Report
Comments and Suggestions for Authors
The authors have responded satisfactorily to the comments made, furthermore, they have specifically mentioned the limitations of their study, which is appreciated. A final figure summarizing and presenting your findings would be very beneficial considering your study model.
